# The Multidimensional Prognostic Index Predicts Mortality in Older Outpatients with Cognitive Decline

**DOI:** 10.3390/jcm11092369

**Published:** 2022-04-23

**Authors:** Femke C. M. S. Overbeek, Jeannette A. Goudzwaard, Judy van Hemmen, Rozemarijn L. van Bruchem-Visser, Janne M. Papma, Harmke A. Polinder-Bos, Francesco U. S. Mattace-Raso

**Affiliations:** 1Department of Geriatric Medicine, Erasmus MC University Medical Center, 3015 GD Rotterdam, The Netherlands; femkeoverbeek@hotmail.com (F.C.M.S.O.); j.goudzwaard@erasmusmc.nl (J.A.G.); r.l.visser@erasmusmc.nl (R.L.v.B.-V.); h.polinder-bos@erasmusmc.nl (H.A.P.-B.); 2Department of Neurology, Erasmus MC University Medical Center, 3015 GD Rotterdam, The Netherlands; j.vanhemmen@erasmusmc.nl (J.v.H.); j.papma@erasmusmc.nl (J.M.P.)

**Keywords:** dementia, Multidimensional Prognostic Index, cognitive decline, aging, mortality, frailty, geriatric assessment

## Abstract

Since the heterogeneity of the growing group of older outpatients with cognitive decline, it is challenging to evaluate survival rates in clinical shared decision making. The primary outcome was to determine whether the Multidimensional Prognostic Index (MPI) predicts mortality, whilst assessing the MPI distribution was considered secondary. This retrospective chart review included 311 outpatients aged ≥65 years and diagnosed with dementia or mild cognitive impairment (MCI). The MPI includes several domains of the comprehensive geriatric assessment (CGA). All characteristics and data to calculate the risk score and mortality data were extracted from administrative information in the database of the Alzheimer’s Center and medical records. The study population (mean age 76.8 years, men = 51.4%) was divided as follows: 34.1% belonged to MPI category 1, 52.1% to MPI category 2 and 13.8% to MPI category 3. Patients with dementia have a higher mean MPI risk score than patients with MCI (0.47 vs. 0.32; *p* < 0.001). The HRs and corresponding 95% CIs for mortality in patients in MPI categories 2 and 3 were 1.67 (0.81–3.45) and 3.80 (1.56–9.24) compared with MPI category 1, respectively. This study shows that the MPI predicts mortality in outpatients with cognitive decline.

## 1. Introduction

Cognitive decline is one of the most significant and common problems in older persons [1]. Dementia is an overall term for chronic and progressive neurodegenerative disorders with the loss of cognitive functioning [2]. Mild cognitive impairment (MCI) is a syndrome of having greater than expected cognitive decline for the person’s age and education level, but in contrast to dementia, it does not significantly interfere with daily life activities [3]. Patients with MCI have an increased risk of developing dementia, and the conversion rate is almost 10% per year [4]. The prevalence rate of dementia in Europe in older persons aged over 65 is about 7%, and for MCI, this is four times greater [5,6]. The number of people living with dementia will double by 2050 [7]. Deaths from dementia are rising, making it the second leading cause of death in high-income countries [8]. Dementia appears to lead to higher mortality and is also associated with a loss of quality of life, higher healthcare costs, and, more often, institutionalization [9,10,11]. Nevertheless, it is still challenging for clinicians to inform their patients based on medical knowledge about the prognosis after being diagnosed with dementia or MCI.

The prognosis of older patients with cognitive decline is determined by their cognitive status and results from a combination of biological, functional, nutritional, environmental, psychological, and socioeconomic factors [12]. The comprehensive geriatric assessment (CGA) is the main tool to integrate information from all these domains and determine the prognosis [13]. In shared clinical decision-making, life expectancy and quality of life are important factors in evaluating diagnostic and therapeutic interventions [14]. Estimating the state of frailty and life expectancy of the patients is essential to point out the most appropriate decisions for treatment, prevention, institutionalization and care management [15]. Considering the heterogeneity of the group of patients with cognitive decline, it is challenging to evaluate survival rates.

The Multidimensional Prognostic Index (MPI) is derived from a standardized CGA. It uses a mathematic algorithm including information on eight domains relevant for assessing frail older persons (i.e., functional, cognitive and nutritional status, cohabitation status, comorbidities, polypharmacy and risk of pressure sores) using standardized and validated rating scales [13]. The MPI generates a numeric score between 0 and 1 that expresses the global risk index of multidimensional impairment. Therefore, the MPI can translate clinical outcomes of the CGA into a risk score that can predict negative outcomes. This tool has been developed and identified as well-calibrated to predict short and long-term mortality in several independent cohorts of hospitalized and community-dwelling older patients with acute and chronic diseases [1,13,14,16,17].

The present study aimed to investigate the distribution and the possible predictive value on mortality of the MPI within a population of outpatients with cognitive decline.

## 2. Materials and Methods

This study was a retrospective chart review study. All outpatients aged 65 years and older who were referred to the Alzheimer Center of the Erasmus MC University Medical Center, Rotterdam, from 1 January 2010 to 31 December 2020, and were primarily seen by a geriatrician were screened for inclusion in the study. Patients diagnosed with dementia or MCI after undergoing the standard workup were included. Exclusion criteria were a diagnosis of frontotemporal dementia (FTD), a psychiatric disorder and if data regarding calculating the MPI risk score was incomplete.

The present study was conducted according to the Declaration of Helsinki. In the Netherlands, ethical approval and patient consent are not required for a retrospective chart review study in which data collected during routine clinical care are extracted and analyzed anonymously. Two physicians were responsible for correctly extracting the information needed (FO, FMR).

All patients were included from the database of the Alzheimer’s Center (*n* = 4461). The following data were already available: age, sex, Mini-Mental State Examination (MMSE) score and diagnosis. Information on mortality and the variables to calculate the MPI risk score were collected from the medical records. The MPI was determined based on information from eight different domains of the CGA [13]:Cohabitation status was divided into three parts: living with family (with spouse and/or other relatives and/or a caregiver), institutionalized and alone;Medication use was defined by the number of drugs used and was ranged into three groups: ≤3, from 4 to 6 and ≥7;Functional status was evaluated by Katz’s Activities of Daily Living (ADL) index [18];Independence was defined by Lawton’s Instrumental Activities of Daily Living (IADL) index [19];Cognitive status was assessed with the Mini-Mental State Examination (MMSE). When appropriate, in case of a diverse cultural background and/or language barrier, the Rowland University Dementia Assessment Scale (RUDAS) was used [20,21];Comorbidity was examined using the Cumulative Illness Rating Scale comorbidity index (CIRS-CI) [22];The risk of developing pressure sores was evaluated through the Exton Smith Scale (ESS) [23];Nutritional status was investigated with the Mini Nutritional Assessment short-form (MNA-SF) [24].

For each domain, a value is determined according to the conventional cutoff points derived from the literature (Table 1). Value 0 indicates no problem, 0.5 is a minor problem, and 1 is a severe problem. The sum of the calculated scores from the eight domains was divided by 8 to obtain the final MPI risk score ranging from 0 = no risk to 1 = highest risk. Additionally, the MPI was expressed as three categories of risk: MPI category 1 low risk (MPI risk score 0–0.33), MPI category 2 moderate risk (MPI risk score 0.34–0.66) and MPI category 3 high risk (MPI risk score 0.67–1) [13].

The diagnosis of dementia was reached within the Alzheimer Center’s team during weekly multidisciplinary consultation, according to the Diagnostic and Statistical Manual of Mental Disorders (DSM) 5th edition criteria [25].

Baseline characteristics were reported as frequencies for categorical variables and mean ± standard deviation (SD) for continuous variables. Comparisons across MPI categories were performed using a Chi-squared test for categorical variables and a one-way ANOVA test for continuous variables. For the comparisons between dementia and MCI, a Chi-squared test was used for categorical variables, and an independent samples T-test was used for continuous variables. To investigate the predictive role of MPI upon mortality, a Cox regression model was used to assess the hazard ratios (HR) and 95% confidence interval (95% CI). The model was adjusted for age and sex. Only 12 patients belonged to the MPI category 3, and we repeated the Cox regression analysis combining MPI categories 2 and 3 for further analysis. All statistical analyses were performed using SPSS Statistics 27. Two-sided *p*-values < 0.05 were considered statistically significant.

## 3. Results

The database included 4461 patients, of which 3802 were not included because of having visited another center, not being primarily seen by a geriatrician or age < 65 years. A total of 659 patients were screened for inclusion, and eventually, the study population included 311 patients (mean age 76.8 ± 6 years, 160 men (51.4%)) (Figure 1).

In total, 73% of patients had dementia, and 27% had MCI. In the dementia group, 45.4% were diagnosed with Alzheimer’s disease, 16.7% with vascular dementia, 18.9% with mixed dementia (Alzheimer’s and vascular dementia) and 18.9% with other forms of dementia (seven patients with Lewy body dementia, two with logopenic progressive aphasia, two with corticobasal degeneration and 32 remained undifferentiated). Table 2 shows the clinical characteristics of the patients included in the study. In total, 106 (34.1%) patients belonged to MPI category 1, 162 (52.1%) to MPI category 2 and 43 (13.8%) to MPI category 3. The majority (66.9%) of the patients lived together with family, 29.9% lived alone, and 3.2% were institutionalized. The daily number of drugs taken was high at 6.1 ± 3.9, and the comorbidity index (CIRS-CI) was rated at 3.2 ± 1.7. The patients had a moderate degree of dependence (ADL 5.3 ± 1.3; IADL 4.8 ± 2.4) but were cognitively impaired according to cognitive screening tests (MMSE 21.8 ± 5; RUDAS 18.8 ± 5.5). The cognitive status was scored using the MMSE in 289 patients (92.9%) and the RUDAS in 22 patients (7.1%). The risk of pressure ulcers was low (ESS score 17.5 ± 2.4), probably since the study population consisted of outpatients. With a mean MNA-SF of 11 ± 2.3, there was a moderate risk of malnutrition.

Table 3 presents the characteristics of the patients according to their MPI category. A higher MPI category is associated with a higher prevalence of dementia and a lower prevalence of MCI (*p* < 0.001 between MPI 1 and MPI 2/3; *p* = 0.008 between MPI 2 and MPI 3). Patients in MPI categories 2 and 3 were older (*p* < 0.05) than patients in MPI category 1. As expected, participants in higher MPI categories were more likely to use more medication, suffer from more comorbidities, be less independent in activities of daily living, have higher pressure sore risk and have a higher risk of malnutrition (*p* < 0.001 in all domains). The cognitive status of patients in MPI categories 2 and 3 was worse than in MPI category 1 (*p* < 0.001). There were fewer men in the dementia group than in the MCI group (45.4% vs. 67.9%, respectively; *p* < 0.001). The mean MPI risk score was higher in patients with dementia than with MCI (0.47 ± 0.18 vs. 0.32 ± 0.15; *p* < 0.001). Patients with dementia had worse ADL and IADL scores, cognitive status according to the MMSE or RUDAS, pressure sore risk and malnutrition risk (*p* < 0.001 in all domains) than patients with MCI. They were also more likely to use more medications than patients with MCI (*p* = 0.035).

Seventy-one patients (22.8%) died during follow-up. After diagnosis, the average patient survival time was 2.5 years (from a minimum of 0.2 to a maximum of 10.4 years). Figure 2 shows the survival curve for the three MPI categories. The HRs and corresponding 95% CIs for mortality in patients in MPI category 2 and MPI category 3 were 1.67 (0.81–3.45) and 3.80 (1.56–9.24), respectively. When we combined patients in MPI categories 2 and 3, the HR for mortality was 3.98 (95% CI: 1.97—8.04, *p* < 0.001) compared to patients in MPI category 1.

## 4. Discussion

In the present study, we found that older outpatients with cognitive decline and high MPI scores have an increased mortality risk. Patients with dementia had higher MPI scores than patients with MCI.

Several prognostic instruments have been investigated in older patients with cognitive decline. Nearly all instruments were validated in specific groups of patients such as institutionalized older patients with advanced dementia and Alzheimer’s disease or investigated only the influence of single characteristics [26,27,28,29]. A systematic review of prognostic instruments based on literature confirmed the lack of prognostic indicators of 6-month mortality in older patients with advanced dementia [30]. A nationwide prospective cohort study in the Netherlands found an increased one- and five-year mortality risk in patients with dementia compared with the general population. Still, specific patient characteristics were not taken into account [31]. Recently, a nationwide registry-linkage study developed a survival time tool for older patients with dementia. This tool appeared to be very accurate in predicting the three-year survival [32].

A large prospective study found that the MPI had significantly better prognostic accuracy than three other frailty indices predicting short- and long-term mortality in different settings in hospitalized older patients [33]. A higher MPI risk score seemed, besides a higher risk of mortality, to be associated with other negative outcomes, such as institutionalization, rehospitalization and access to home care services [16,34]. As expected and in agreement with other recent studies, the MPI was also effective in predicting mortality in older patients with cognitive decline in this study [1,34]. The study of Pilotto et al. was based on hospitalized patients with dementia, and MCI was not considered. However, no study had explored whether the prevalence of dementia and MCI was higher or lower among the MPI categories [1].

The findings of our study should be interpreted with some limitations. First, the patients included were recruited within a single university medical center, and therefore we cannot extrapolate our findings to a general population. Second, we have used a modified version of the original MPI. Nevertheless, the different tools used in the present study were previously validated [22,35,36]. Third, transitions between the MPI categories over time can occur, potentially affecting the results. Information on education level was not available. Therefore, we cannot exclude that this might have biased the results.

This study has several strengths. First, a relatively large number of outpatients was included. Second, since the diagnosis is based on multidisciplinary consultation using biomarkers, diagnostic imaging and neuropsychological assessment, it is less likely that the diagnosis of dementia or MCI is missed or misdiagnosed. Third, including patients when only the RUDAS was available contributed to a culturally diverse background study population.

In conclusion, we found that a high MPI risk score was associated with an increased mortality rate in a group of older outpatients with cognitive decline. These findings need to be confirmed in larger and heterogeneous populations of patients with cognitive decline. If confirmed, the MPI would be a novel tool for risk stratification and medical decisions in this peculiar category of patients.

## Figures and Tables

**Figure 1 jcm-11-02369-f001:**
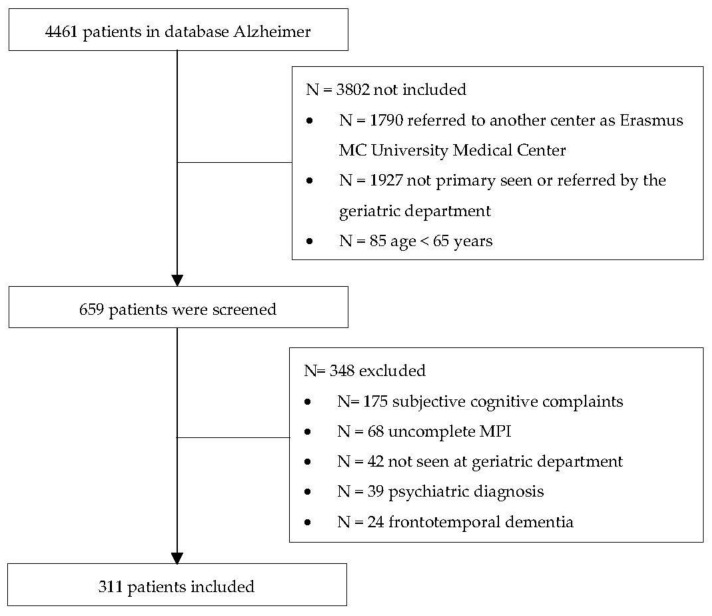
Flow chart of the study.

**Figure 2 jcm-11-02369-f002:**
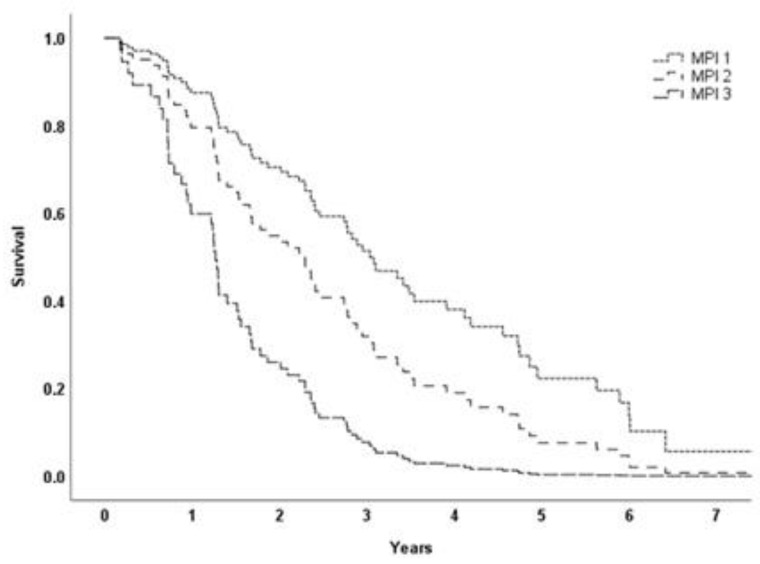
Cox regression survival curve stratified by MPI groups. Model adjusted for age and sex.

**Table 1 jcm-11-02369-t001:** MPI risk score is assigned to each domain based on the severity of the problem.

Assessment	No Problem (Value = 0)	Minor Problem (Value = 0.5)	Severe Problem (Value = 1)
CHS	Living with family	Institutionalized	Alone
Number of medications	0–3	4–6	≥7
ADL	6–5	4–3	2–0
IADL	8–6	5–4	3–0
MMSE (*n* = 289) RUDAS (*n* = 22)	28–30 26–30	25–27 17–25	0–24 0–16
CIRS-CI	0	1–2	≥3
ESS	16–20	10–15	5–9
MNA-SF	≥12	8–11	0–7

Notes: Values are given in points. Abbreviations: MPI, Multidimensional Prognostic Index; CHS, cohabitation status; ADL, Activities of Daily Living; IADL, Instrumental Activities of Daily Living; MMSE, Mini-Mental State Examination; RUDAS, Rowland Universal Dementia Assessment Scale; CIRS-CI, Cumulative Illness Rating Scale Comorbidity Index; ESS, Exton Smith Scale; MNA-SF, Multi Nutritional Assessment Short Form.

**Table 2 jcm-11-02369-t002:** General characteristics (*n* = 311).

Men	160
Age (years)	76.8 ± 6
MCI	84
Dementia	227
Alzheimer’s disease	103
Vascular dementia	38
Mixed dementia	43
Other	43
MPI-1	106
MPI-2	162
MPI-3	43
MPI risk score	0.43 ± 0.19
CHS	
With family	208
Institutionalized	10
Alone	93
Number of medications	6.1 ± 3.9
ADL	5.3 ± 1.3
IADL	4.8 ± 2.4

Notes: Values are given as n or mean ± SD. Abbreviations: SD, standard deviation; MCI, mild cognitive impairment; MPI, Multidimensional Prognostic Index; CHS, cohabitation status; ADL, activities of daily living; IADL, instrumental activities of daily living; MMSE, Mini-Mental State Examination; RUDAS, Rowland Universal Dementia Assessment Scale; CIRS-CI, Cumulative Illness Rating Scale Comorbidity Index; ESS, Exton Smith Scale; MNA-SF, Multi Nutritional Assessment Short Form.

**Table 3 jcm-11-02369-t003:** General characteristics according to MPI categories.

Variable	MPI 1 (*n* = 106)	MPI 2 (*n* = 162)	MPI 3 (*n* = 43)	*p*-Value
Men	56	88	16	* ns
				** ns
				*** 0.046
Age (years)	75.6 ± 5.8	77.1 ± 5.7	78.3 ± 6.9	* 0.045
				** 0.012
				*** ns
Dementia	58	127	42	* <0.001
MCI	48	35	1	** <0.001
				*** 0.008
CHS				
With family	88	105	15	
Institutionalized	0	5	5	
Alone	18	52	23	
Number of medications	3.3 ± 3.1	7 ± 3.3	9.9 ± 3.3	<0.001
ADL	5.9 ± 0.4	5.5 ± 0.9	3.1 ± 1.6	<0.001
IADL	6.5 ± 1.7	4.4 ± 2.2	2 ± 1.6	<0.001
MMSE	24.5 ± 3.9	20.7 ± 5.1	19.1 ± 4.4	* <0.001
				** <0.001
				*** ns
RUDAS	21.8 ± 5	18.3 ± 5.5	17.2 ± 6.1	ns
CIRS-CI	1.9 ± 1.3	3.6 ± 1.4	4.8 ± 1.4	<0.001

Notes: Values are given as n or mean ± SD. Abbreviations: SD, standard deviation; ns, not significant; MCI, mild cognitive impairment; MPI, Multidimensional Prognostic Index; CHS, cohabitation status; ADL, activities of daily living; IADL, instrumental activities of daily living; MMSE, Mini-Mental State Examination; RUDAS, Rowland Universal Dementia Assessment Scale; CIRS-CI, Cumulative Illness Rating Scale Comorbidity Index; ESS, Exton Smith Scale; MNA-SF, Multi Nutritional Assessment Short Form. * *p*-value between MPI 1 and MPI 2. ** *p*-value between MPI 1 and MPI 3. *** *p*-value between MPI 2 and MPI 3.

## Data Availability

Database of the Alzheimer Center of the Erasmus MC University Medical Center, Rotterdam.

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
