# Peer review of "The Multidimensional Prognostic Index Predicts Mortality in Older Outpatients with Cognitive Decline"

_jcm, 2022, doi:10.3390/jcm11092369_

Round 1

Reviewer 1 Report

Dear Authors:

Could you please clarify the underlying reasons why high MPI may cause higher mortality of MCI patients? Thank you.

Best,

Author Response

Dear Reviewer 1,

Thank you very much for having read and evaluated the manuscript The Multidimensional Prognostic Index predicts mortality in older outpatients with cognitive decline (jcm-1649455).

We appreciate your positive response and want to address your observation “Could you please clarify the underlying reasons why high MPI may cause higher mortality of MCI”.

An increasing body of evidences indicates that the prognosis of older patients is strongly related to the presence of concomitant diseases and to the degree of physical, cognitive, biological, and social impairment. The Comprehensive Geriatric Assessment (CGA) is the tool of choice to effectively explore these domains of health, especially in the functionally compromised and frail older subject. The Multidimensional Prognostic Index (MPI) is a product of the CGA, that uses a mathematic algorithm including information about eight domains relevant for the global assessment of the older persons (i.e., functional and cognitive status, nutrition, mobility and risk of pressure sores, multimorbidity, polypharmacy and co-habitation), to generate a numeric score (or index), between 0 and 1, that expresses the global risk of multidimensional impairment. In this sense, we can say that MPI is able to translate the clinical evaluation of the CGA in a score that can accurately predict overall mortality and other negative outcomes, and therefore, be used as prognostic tool in older people. We have added more information on this in the introduction, please see page 2, lines 51-61. Furthermore, typos were corrected where appropriate.

Reviewer 2 Report

This is an interesting study examining the predictive value of the Multidimensional Prognostic Index within a population of outpatients with cognitive decline. The paper is promising and may provide important contribution to the literature. It is also well-written. I have a few comments to improve the manuscript further:

  1. In the introduction, it will be important for the authors to elaborate and provide more details about the Multidimensional Prognostic Index. There should also be more information and justification on each dimension in the MPI.
  2. It will be important for the authors to take into account participants' education level in their study. Education level should be controlled in the analyses. This should be noted in the limitation if the variable is not available.
  3. It will be informative for the authors to report whether the assumptions in the Cox model were not violated in the analysis.

Author Response

Dear Reviewer 2,

Thank you very much for having read and evaluated the manuscript The Multidimensional Prognostic Index predicts mortality in older outpatients with cognitive decline (jcm-1649455).

We appreciate your positive response and want to address your observations.

Reviewer 2.

This is an interesting study examining the predictive value of the Multidimensional Prognostic Index within a population of outpatients with cognitive decline. The paper is promising and may provide important contribution to the literature. It is also well-written. I have a few comments to improve the manuscript further:

Overbeek et al.

Thank you very much for your compliments and typos were corrected where appropriate.

Reviewer 2.

In the introduction, it will be important for the authors to elaborate and provide more details about the Multidimensional Prognostic Index. There should also be more information and justification on each dimension in the MPI.

Overbeek et al.

An increasing body of evidences indicates that the prognosis of older patients is strongly related to the presence of concomitant diseases and to the degree of physical, cognitive, biological, and social impairment. The Comprehensive Geriatric Assessment (CGA) is the tool of choice to effectively explore these domains of health, especially in the functionally compromised and frail older subject. The Multidimensional Prognostic Index (MPI) is a product of the CGA, that uses a mathematic algorithm including information about eight domains relevant for the global assessment of the older persons (i.e., functional and cognitive status, nutrition, mobility and risk of pressure sores, multimorbidity, polypharmacy and co-habitation), to generate a numeric score (or index), between 0 and 1, that expresses the global risk of multidimensional impairment. In this sense, we can say that MPI is able to translate the clinical evaluation of the CGA in a score that can accurately predict overall mortality and other negative outcomes, and therefore, be used as prognostic tool in older people. We have added more information on this in the introduction, please see page 2, lines 51-61.

Reviewer 2.

It will be important for the authors to take into account participants' education level in their study. Education level should be controlled in the analyses. This should be noted in the limitation if the variable is not available.

Overbeek et al.

We really appreciate this suggestion. We are aware that education level can have effect on cognitive functions as well on outcomes. Unfortunately we did not have this information, therefore we are not able to perform analyses adjusted for educational level. Since we agree with the reviewer that this is relevant information we address this issue in the discussion when addressing the limitations, please see page 8, lines 237-239.

Reviewer 2. It will be informative for the authors to report whether the assumptions in the Cox model were not violated in the analysis.

Overbeek et al.

The assumptions in the Cox model are not violated in the analysis, since the survival curves for the different MPI categories have hazard functions that are proportional over the time in years and secondly since the relationship between the log hazard and each covariate is linear.

Round 2

Reviewer 1 Report

Strongly suggest for publishing.

Reviewer 2 Report

The authors have sufficiently addressed my concerns. I appreciate their effort.